# Topological data analysis reveals a core gene expression backbone that defines form and function across flowering plants

Sourabh Palande[1], Joshua A. M. Kaste[2,3], Miles D. Roberts[3], Kenia Segura Abá[3], Carly Claucherty[4], Jamell Dacon[5], Rei Doko[5], Thilani B. Jayakody[4], Hannah R. Jeffery[4], Nathan Kelly[6], Andriana Manousidaki[7], Hannah M. Parks[2], Emily M. Roggenkamp[4], Ally M. Schumacher[3], Jiaxin Yang[1], Sarah Percival[2], Jeremy Pardo[3], Aman Y. Husbands[8], Arjun Krishnan[1,2], Beronda L Montgomery[2,9,10], Elizabeth Munch[1,11], Addie M. Thompson[4,12], Alejandra Rougon-Cardoso[13,14], Daniel H. Chitwood[1,6]*, Robert VanBuren[6,12]*

1 Department of Computational Mathematics, Science & Engineering, Michigan State University, East Lansing, Michigan, United States of America, 2 Department of Biochemistry and Molecular Biology, Michigan State University, East Lansing, Michigan, United States of America, 3 Department of Plant Biology, Michigan State University, East Lansing, Michigan, United States of America, 4 Department of Plant, Soil & Microbial Sciences, Michigan State University, East Lansing, Michigan, United States of America, 5 Department of Computer Science and Engineering, Michigan State University, East Lansing, Michigan, United States of America, 6 Department of Horticulture, Michigan State University, East Lansing, Michigan, United States of America, 7 Department of Statistics and Probability, Michigan State University, East Lansing, Michigan, United States of America, 8 Department of Biology, University of Pennsylvania, Philadelphia, Pennsylvania, United States of America, 9 Department of Microbiology & Molecular Genetics, Michigan State University, East Lansing, Michigan, United States of America, 10 MSU-DOE Plant Research Laboratory, Michigan State University, East Lansing, Michigan, United States of America, 11 Department of Mathematics, Michigan State University, East Lansing, Michigan, United States of America, 12 Plant Resilience Institute, Michigan State University, East Lansing, Michigan, United States of America, 13 Laboratory of Agrigenomic Sciences, Universidad Nacional Autónoma de México, ENES-León, León, Mexico, 14 Laboratorio Nacional Plantecc, ENES-León, León, Mexico

☙ These authors contributed equally to this work.
* dhchitwood@gmail.com (DHC); bobvanburen@gmail.com (RV)

**Data Availability Statement:** The code, metadata, and raw datasets from this project are available on a dedicated GitHub page: https://github.com/

## Abstract

Since they emerged approximately 125 million years ago, flowering plants have evolved to dominate the terrestrial landscape and survive in the most inhospitable environments on earth. At their core, these adaptations have been shaped by changes in numerous, interconnected pathways and genes that collectively give rise to emergent biological phenomena. Linking gene expression to morphological outcomes remains a grand challenge in biology, and new approaches are needed to begin to address this gap. Here, we implemented topological data analysis (TDA) to summarize the high dimensionality and noisiness of gene expression data using lens functions that delineate plant tissue and stress responses. Using this framework, we created a topological representation of the shape of gene expression across plant evolution, development, and environment for the phylogenetically diverse flowering plants. The TDA-based Mapper graphs form a well-defined gradient of tissues from leaves to seeds, or from healthy to stressed samples, depending on the lens function. This suggests that there are distinct and conserved expression patterns across angiosperms that delineate different tissue types or responses to biotic and abiotic stresses. Genes that

PlantsAndPython/plant-evo-mapper and Zenodo:
https://zenodo.org/records/8428609

**Funding:** This work was funded primarily by National Science Foundation Research Traineeship training grant (NSF 1828149 to ATM, DHC, and RV) which established the Integrated training Model in Plant And Compu-Tational Sciences (IMPACTS) program at Michigan State University. This grant funded fellows within this program (JAMK, MDR, KSA, CC, JD, RD, TBJ, HRJ, AM, EMR, AMS, JY) as well as the project-based curriculum for the Plants and Python Course that formed the backbone of this manuscript. This work is also supported by NSF Plant Genome Research Program awards IOS-2310355 to EM, DHC, and RV, IOS-2310356 to AH, and IOS-2310357 to AK, NSF Developmental Mechanisms award IOS-2039489 to AH, and NSF Biological Integration Institute award (DBI-2213983 to RV). Several students (JAMK, MDR, KSA, HMP, JP) were supported by predoctoral training award (T32-GM110523 to RV) from the National Institute of General Medical Sciences of the NIH. This project was supported by the USDA National Institute of Food and Agriculture, and by Michigan State University AgBioResearch to AMT, DHC, and RV. The funders had no role in study design, data collection and analysis, decision to publish, or preparation of the manuscript.

**Competing interests:** The authors have declared that no competing interests exist.

**Abbreviations:** GO, gene ontology; PCA, principal component analysis; SRA, sequence read archive; SVA, surrogate variable analysis; TDA, topological data analysis; TPM, transcript per million; t-SNE, t-distributed stochastic neighbor embedding.

correlate with the tissue lens function are enriched in central processes such as photosynthetic, growth and development, housekeeping, or stress responses. Together, our results highlight the power of TDA for analyzing complex biological data and reveal a core expression backbone that defines plant form and function.

## Introduction

Over 300,000 gene expression datasets have been collected for thousands of diverse plant species spanning over 900 million years of divergence [1]. This wealth of publicly available datasets spans ecological niches, species, developmental stages, tissues, stresses, and even single cells, providing a largely untapped reservoir of biological information. These diverse datasets provide an opportunity to link insights from various biological disciplines, including ecology, development, physiology, genetics, evolution, biochemistry, and cell biology through a common computational and mathematical framework. These gene expression datasets have been analyzed individually for specific experiments and hypotheses, but large-scale meta-analyses across the publicly available expression datasets are largely nonexistent for plants.

Beyond a common currency that links the subdisciplines of biology, gene expression links its emergent levels. Below gene expression, the genome gives rise to transcriptional networks and protein interactions that are directly responsible for the complexity of gene expression. Above it, gene expression orchestrates cell-specific expression and the development of the organism itself, impacting phenotypes ranging from physiology to plasticity that propagate further to the population, community, and ecological levels. These features, from molecular (DNA, promoter sequences, -omics datasets) to the organismal, population, and ecological levels (life history traits, climatic data from species distributions, etc.) have been used in the past as labels and predicted outputs of machine learning models [2,3]. The structure—the shape—of gene expression in flowering plants is therefore a constraint that is formed by and impacts biological phenomena below and above it, respectively.

Data visualization lies at the heart of exploratory data analysis and provides us with a powerful tool for generating hypotheses that can later be examined using standard statistical techniques. In the era of Big Data, the development of new data visualization pipelines has become increasingly important due to the high dimensionality of the datasets generated and the need to identify patterns and structures that can then become targets for more focused studies. Just as we can look upon the shape of a leaf and derive insights into how it functions from multiple perspectives (developmental, physiological, and evolutionary), we can visualize the shape of any type of data using a Mapper graph [4]. The Mapper algorithm takes as input a filter function that describes a biological aspect of the data and uses mathematical ideas of shape to return a graph that reveals the underlying structure of the data. Even abstract data types like gene expression datasets, therefore, have a shape that we can visualize and derive insights from. For example, Nicolau and colleagues visualized the structure of breast cancer gene expression, identifying 2 distinct branches with differing underlying genotypes and prognostic outcomes that traditional statistical and bioinformatic approaches fail to resolve [5]. This structure was revealed using a pairwise correlation distance matrix as input and modeling of the residuals of each sample from a vector of healthy gene expression as a measure of disease severity. In a second example, using a lens of developmental stage on single-cell RNASeq data, Rizvi and colleagues visualized the underlying structure of gene expression during murine embryonic stem cell differentiation, revealing transient states as well as asynchronous and

continuous transitions between cell types [6]. In both examples, Mapper allowed the shape of data, through a selected lens, to be visualized. The resulting topology of the graph—in the form of loops, branch points, or flares—allowed previously hidden structures to be seen and novel insights to be derived. Loops, branch points, and flares in topological data analysis (TDA)-based Mapper graphs are visual representations of patterns, transitions, and outliers in the data. They provide insights into the topological structure and organization of the data, helping to identify clusters, subgroups, and potential anomalies. Loops represent recurring patterns or relationships in the data, branch points occur when different subsets of data points exhibit distinct topological characteristics, and flares typically indicate outliers or subgroups within a larger cluster and can help identify regions of interest or anomalous behavior in the data.

Surveys of gene expression capture tens of thousands of data points per sample, and this high dimensionality can be represented by a unique shape that underlies emergent biological features. This shape explains gene expression along evolutionary, developmental, and environmental trajectories, leading to innovations that have marked the successful adaptation and proliferation of plant species. To visualize this shape is to better understand what transcriptional profiles are possible and to know the boundaries or constraints that permit or limit gene expression. Here, we analyzed publicly available gene expression profiles across diverse flowering plant families and visualized the underlying structure of gene expression in plants as a graph using the Mapper algorithm. We identified unique topological shapes of plant gene expression when viewed through lenses that delineate different tissue or stress responses. These complex, emergent patterns were largely hidden by biological complexity and sample heterogeneity. Our results demonstrate the ability of Mapper to uncover these patterns in high-dimensional plant gene expression datasets and its potential as a powerful tool for biological hypothesis generation.

## Results

### A representative catalog of flowering plant gene expression

The vast number of gene expression datasets in plants provides a unique opportunity to search for patterns of conservation and divergence throughout angiosperm evolution, across developmental time, tissues, and stress response axes. Previous studies have tried to find common signatures that define different plant tissues or responses to abiotic/biotic stresses, but these have been limited in species breadth [7], depth [8], or had limited downstream analyses [9]. Here, we reanalyzed public expression data on the NCBI sequence read archive (SRA) and applied a topological data analysis method to map the shape of gene expression in plants. We included 54 species that captured the broadest phylogenetic diversity within angiosperms while maximizing the breadth of expression at the tissue and stress levels (Fig 1A). This includes 44 eudicots across 13 families and 9 monocot species across 2 families, as well as *Amborella trichocarpa*, which is sister to the rest of angiosperms. Raw reads were downloaded, cleaned, and reprocessed through a common RNAseq pipeline to remove artifacts related to the different algorithms and downstream analyses used by each group. After filtering datasets with low read mapping, our final set of expression data includes 2,671 samples across 7 distinct developmental tissues and 9 stress classifications for 54 species.

To facilitate comparisons of gene expression across species, we limited our analysis to a set of 6,328 orthologous low-copy genes that were conserved across all 54 plant species using Orthofinder [10]. These sets of orthologous genes or orthogroups are mostly single copy in our diploid species and scale with ploidy in polyploid species. The orthogroups are conserved across a diverse selection of Angiosperm lineages and correspond to well-conserved biological

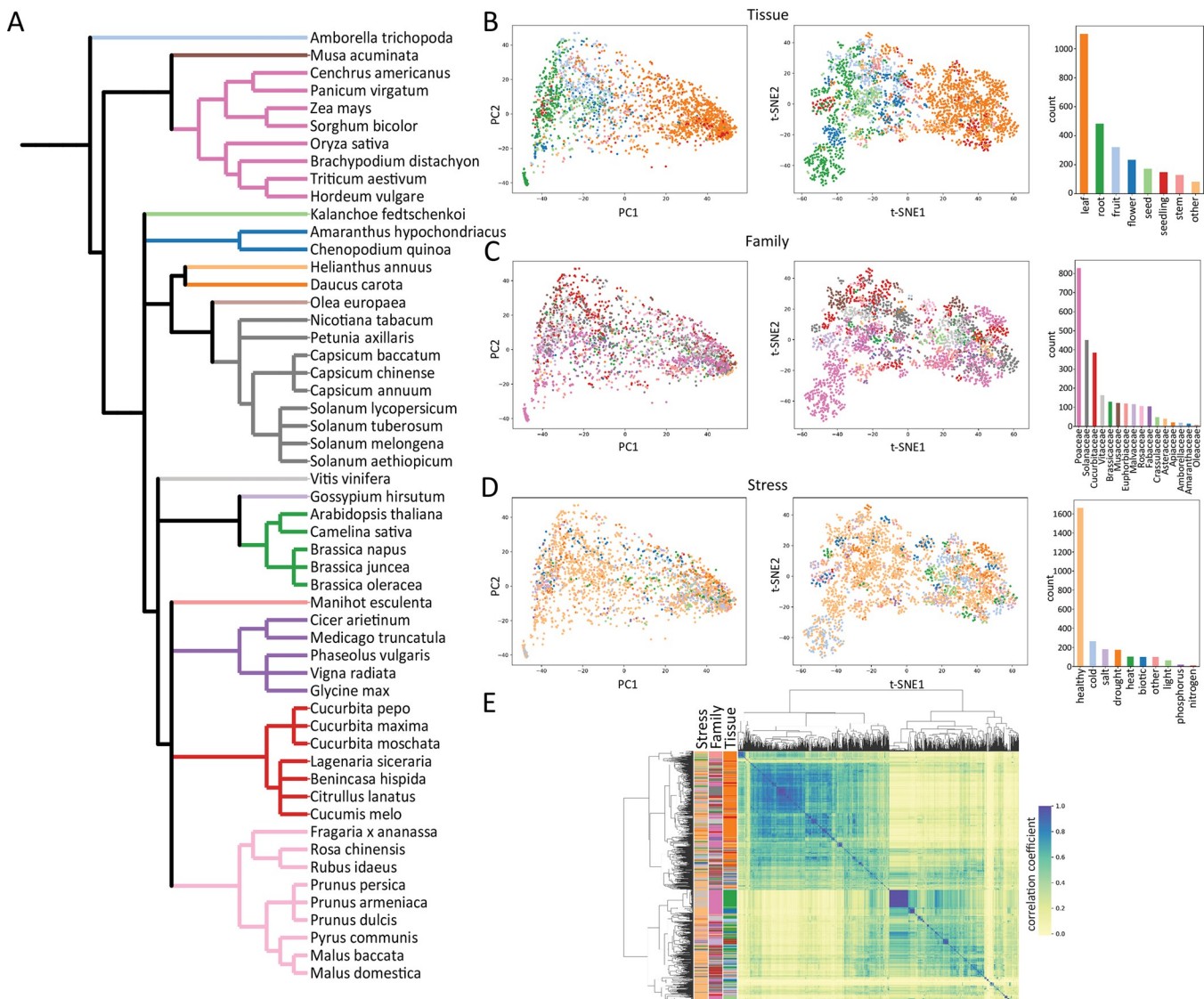

**Fig 1. Dimensional space of plant gene expression across evolution, development, and stress.** (**A**) Representative phylogeny of the 54 plant species included in this study. Nodes (species) are colored by plant family as denoted in Fig 1C. Dimensionality reduction of all samples by principal components (left) and t-SNE (right) are shown for tissue type (**B**), plant family (**C**), and abiotic/biotic stress (**D**). Individual samples are quantified and colored by tissue, family, and stress as shown in the respective bar plots. (**E**) Hierarchical clustering of samples with various biological features highlighted (stress, family, and tissue). Raw expression data underlying the graphs in this figure can be found in S7 Dataset, and code to regenerate analyses can be found in https://zenodo.org/records/8428609 [65].

processes. Gene ontology (GO) term enrichment analysis on the *Arabidopsis thaliana* loci associated with these orthogroups show enrichment for basic biological functions like "DNA replication initiation" and "tRNA methylation" at the top of the list of enriched GO terms, as well as functions specific to photosynthetic organisms like "photosystem II assembly," and "tetraterpenoid metabolic process." Although the remaining orthogroups contain significant biological information, they were excluded from analysis as multigene families typically have diverse functions with divergent expression profiles that would conflate downstream comparative analyses.

The transcript per million (TPM) counts were summed for all genes within an orthogroup for a given species and merged into a single dataframe to create a final matrix of 6,335 orthologs by 2,671 samples. Principal component analysis (PCA) [11] and t-distributed stochastic neighbor embedding (t-SNE) [12] based dimensionality reduction show some separation of samples by different biological factors (Fig 1). The sample space is most clearly delineated by tissue, where both PC1 (explaining 25.4% variation) and t-SNE1 separate the samples into a gradient from root to leaf tissues with other plant tissues sandwiched in between (Fig 1B and 1D). This distribution largely correlates with tissue function, as the sink tissues of flowers, seeds, and fruits resolve closer to the root samples along t-SNE1 and PC1. No tissue type is separated fully by either dimensionality reduction approach. Samples from the 16 plant families are distributed throughout the dimensional space, suggesting that family- or species-level traits are not masking emergent features of distinct tissues (Fig 1C). Interestingly, abiotic and biotic stresses are similarly distributed throughout the dimensional space, with no clear grouping of the same stress across species or individual experiments. This could be due to intrinsic differences in how individual species respond to stress or to differences in the way stress experiments are carried out by different research groups. To account for batch effects and the influence of unmodeled factors, we applied surrogate variable analysis (SVA) to generate estimates of surrogate variables and their effects on our expression matrices. We identified 24 surrogate variables within the dataset, but these latent variables were intrinsically linked to the primary factors in our study (e.g., stress, tissue, and family). Removing surrogate variables would have masked much of the biology we were attempting to quantify, so we chose not to use these "data cleaning" approaches (see Text A in S1 Text for more details).

## Topological data analysis and the shape of plant gene expression

Traditional dimensionality reduction and hierarchical clustering provided some degree of separation, but they were unable to delineate samples by stress or to identify expression patterns related to biological function. This may be related to residual heterogeneity, noise, or because of the inherent biological complexity that underlies plant evolution and function. To test these possibilities, we used a topological data analysis approach to map the shape of our data. TDA was implemented using Mapper [13], which provides a compact, multiscale representation of the data that is well suited for visual exploration and analysis. Mapper is particularly well suited for genomics data as these datasets typically have extremely high dimensionality and sparsity [5]. To construct mapper graphs from our gene expression data, we created 2 different lenses of tissue and stress, adopting an approach similar to Nicolau and colleagues' (Fig 2A–2E). To create the stress lens, we first identified all the healthy samples from the dataset and fit a linear model to them (Fig 2; see Methods). This model serves as the idealized healthy orthogroup expression. We then projected all the samples onto this linear model and obtained the residuals. These residuals measure the deviation of the sample gene expression from the modeled healthy expression, and the lens function is simply the length of the residual vector.

The obvious separation between leaf and root samples in the dimension reduction plots supports a strong photosynthetic versus nonphotosynthetic divide. We used this observation to create a binary tissue lens in the same way as the stress lens. We identified all the photosynthetic samples (i.e., leaf tissue) and created an idealized expression profile by fitting a linear model to these expression profiles (Fig 2). We then projected all the samples onto this linear model and obtained the residuals to establish the lens function by tissue. To define the cover for each lens, we divided the range of the lens function into intervals of uniform length, with the same amount of overlap between adjacent intervals. We experimented with a range of value lengths of the intervals and the size of the overlap to identify the values that produced

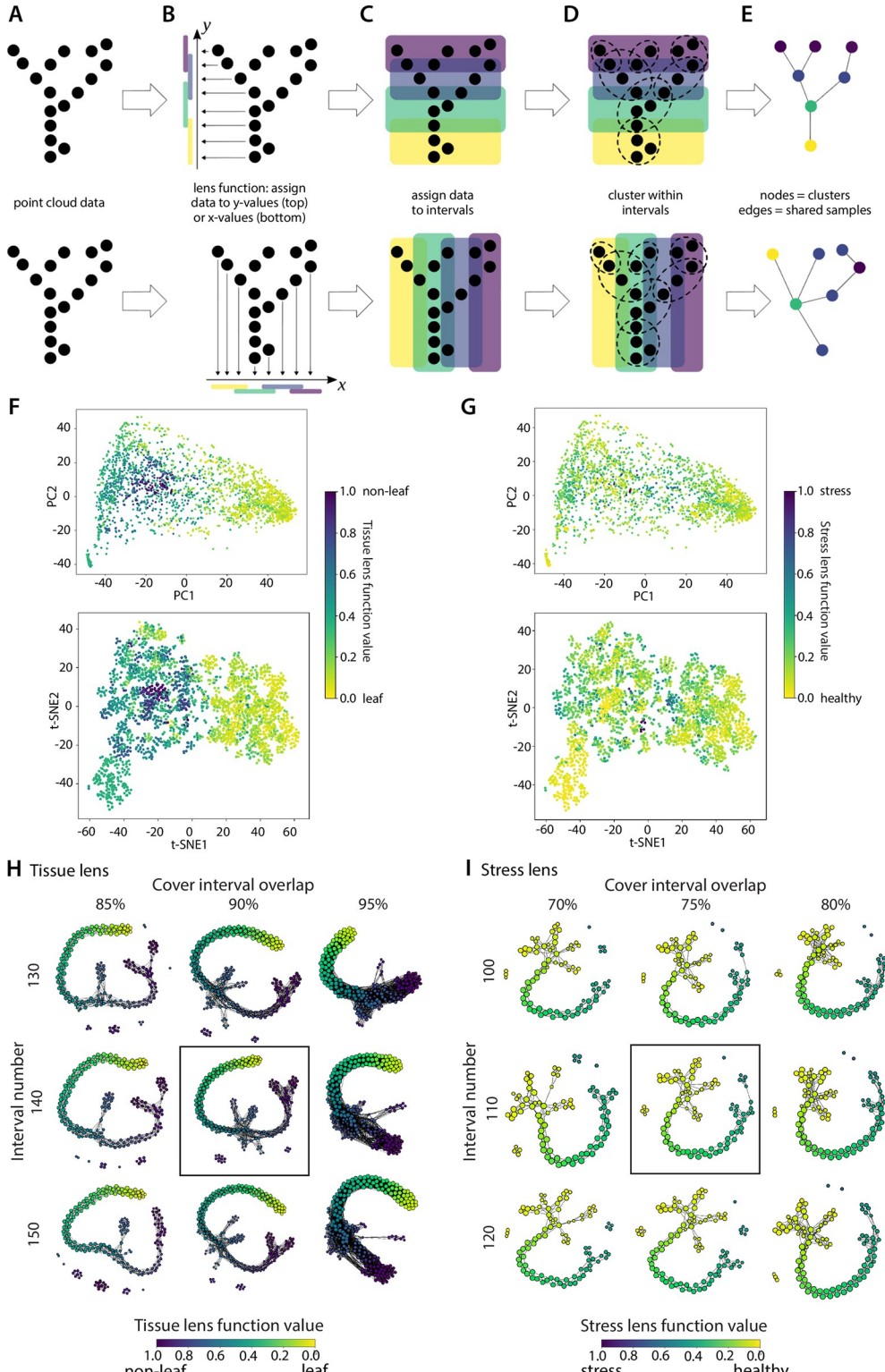

**Fig 2. Topology-based Mapper graphs and the shape of gene expression in plants.** Overview of Mapper graph construction and lens functions (**A**-**E**). The lens function value of each sample is shown in the principal component (top) and t-SNE (bottom) based dimensional reduction from Fig 1 for the tissue (**F**) and stress lens (**G**). Mapper graphs across variable cover intervals and interval number for the tissue (**H**) and stress (**I**) lens function. The Mapper graph constructions we chose for further analysis are enclosed within a box. Raw expression data underlying the graphs in

this figure can be found in S7 Dataset, and code to regenerate analyses can be found in https://zenodo.org/records/8428609 [65].

relatively stable mapper graphs. The clustering was performed using DBSCAN, a commonly used clustering algorithm in Mapper [14].

Overlaying the tissue lens value of each sample over the PCA and t-SNE dimensional space reveals a clear gradient across PC1 and t-SNE1, with the highest lens function values found in seed, fruit, and flower tissues (Fig 2F). For the stress lens function, samples are distributed across the dimensional space, with no obvious correlation between healthy and stressed lens values, similar to the observation from individual abiotic/biotic stresses (Figs 1D and 2G).

Mapper graphs for the tissue and lens functions reflect an emergent and striking topological shape of plant expression (Fig 2H and 2I). Each node in the Mapper graphs corresponds to a bin of similar RNAseq samples with color representing the average lens value of samples within each node. Edges (connections) show common samples between overlapping bins. Changing the cover interval overlap and interval number has marginal effects on the core graph structure but changes the shape and connectivity of sparse nodes on the outskirts of the graphs (Fig 2H and 2I). This central stability highlights the robustness of our input data and significance of the underlying features defining the graph shape [15]. The Mapper graphs for both the tissue and stress lens functions show a backbone structure with numerous embedded nodes and flares that form a well-defined gradient from leaf to seed or healthy to stressed, respectively. This suggests that there are distinct and conserved expression patterns across angiosperms that delineate different tissues or responses to biotic and abiotic stresses.

Our input dataset is unbalanced, with large discrepancies in the number of input samples for different species, stresses, or tissue types. We tested if biases in the distribution of samples could explain the topological shape we observed. We downsampled the most frequent factor combinations and surveyed the effect it had on the Mapper graph topology. Our study has 3 factors: family, tissue, and stress with 16 families, 8 tissue types, and 10 stresses. In total, 1,280 unique 3-way combinations are possible (family + tissue + stress), but in our dataset, only 195 unique combinations are present and they have a heavily skewed distribution (Fig A in S1 Text). Based on this distribution, we chose a cutoff of 30 and downsampled the 30 most common factor combinations. This significantly reduced the sampling bias for family, tissue, and stress, but it did not eliminate them (Fig B in S1 Text). We then reran the Mapper algorithm using this downsampled dataset. The topology is quite similar, suggesting that biases in sample representation are not the major factor underlying the patterns we observed (Fig C in S1 Text).

## Topological shape reflects the underlying biological features of gene expression

To identify and characterize these conserved biological patterns, we first simplified the Mapper graphs into 18 nodes for both the tissue and stress lens functions (Figs 3 and 4). The core tissue-based Mapper graph has discrete nodes for each surveyed plant tissue with a gradual transition of leaves (node 1), to roots (2), fruits (11 and 13), and, finally, seeds (14, 15, and 16; Fig 3A). At the fourth node, the Mapper graph proliferates into terminal branches of flower (node 9), stem (10), fruit (12), and mixtures of uncategorized tissue types (5 and 8). RNAseq samples from the 16 angiosperm families are largely dispersed across nodes by tissue, with some notable exceptions (Fig 3B). Most fruit samples are found along the gradient of the core graph structure, but fruits from the rose (Rosaceae) family form a separate node (node 12). Flowers from the eudicot species are mixed with fruit tissues in nodes along the core graph structure,

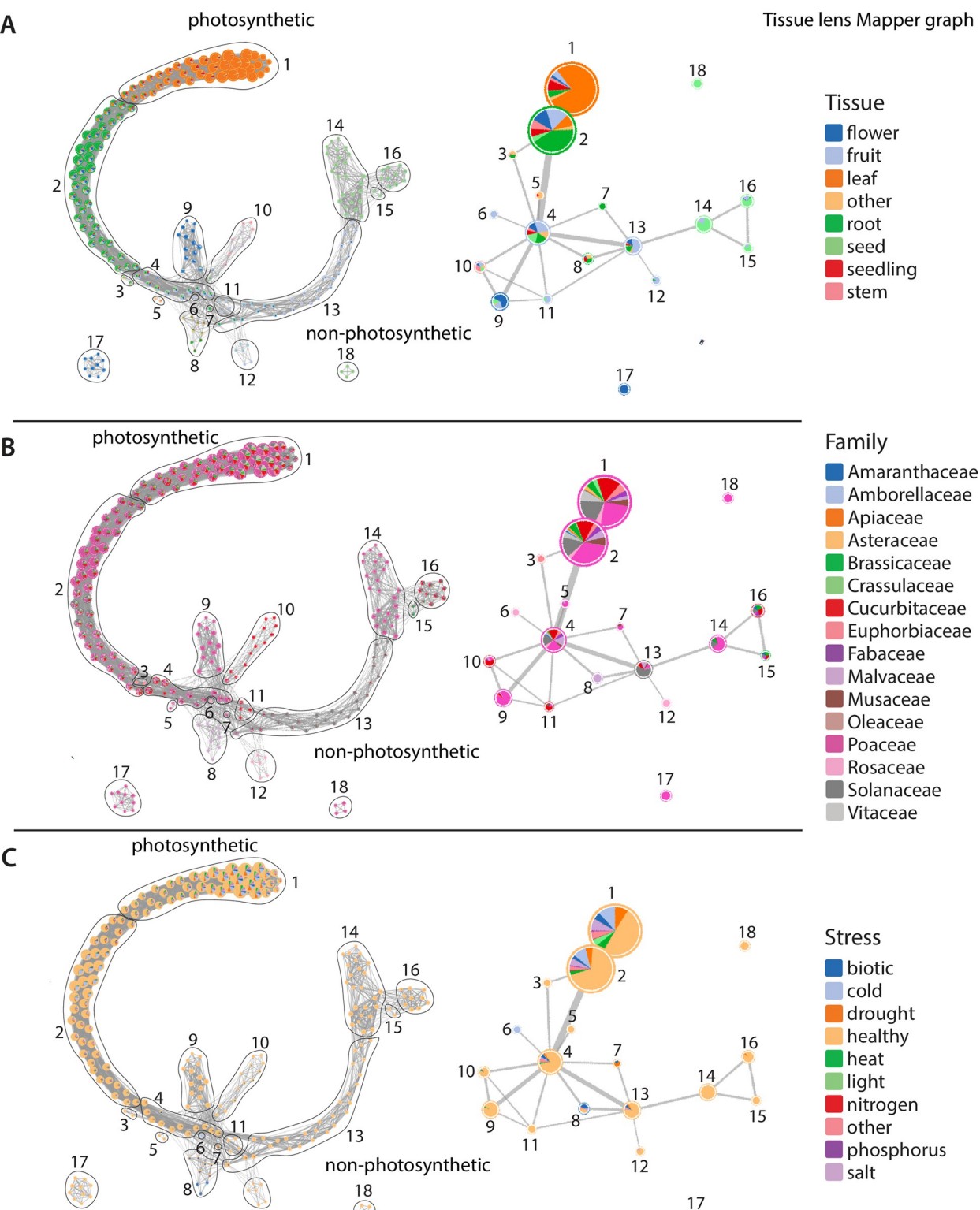

**Fig 3. Simplified Mapper graphs detailing the distribution of samples along the tissue lens.** Nodes along the full Mapper graphs (left) are clustered into simplified Mapper graphs (right), and samples are colored by tissue (**A**), family (**B**), and stress category (**C**). Photosynthetic and nonphotosynthetic ends of the Mapper graph are indicated.

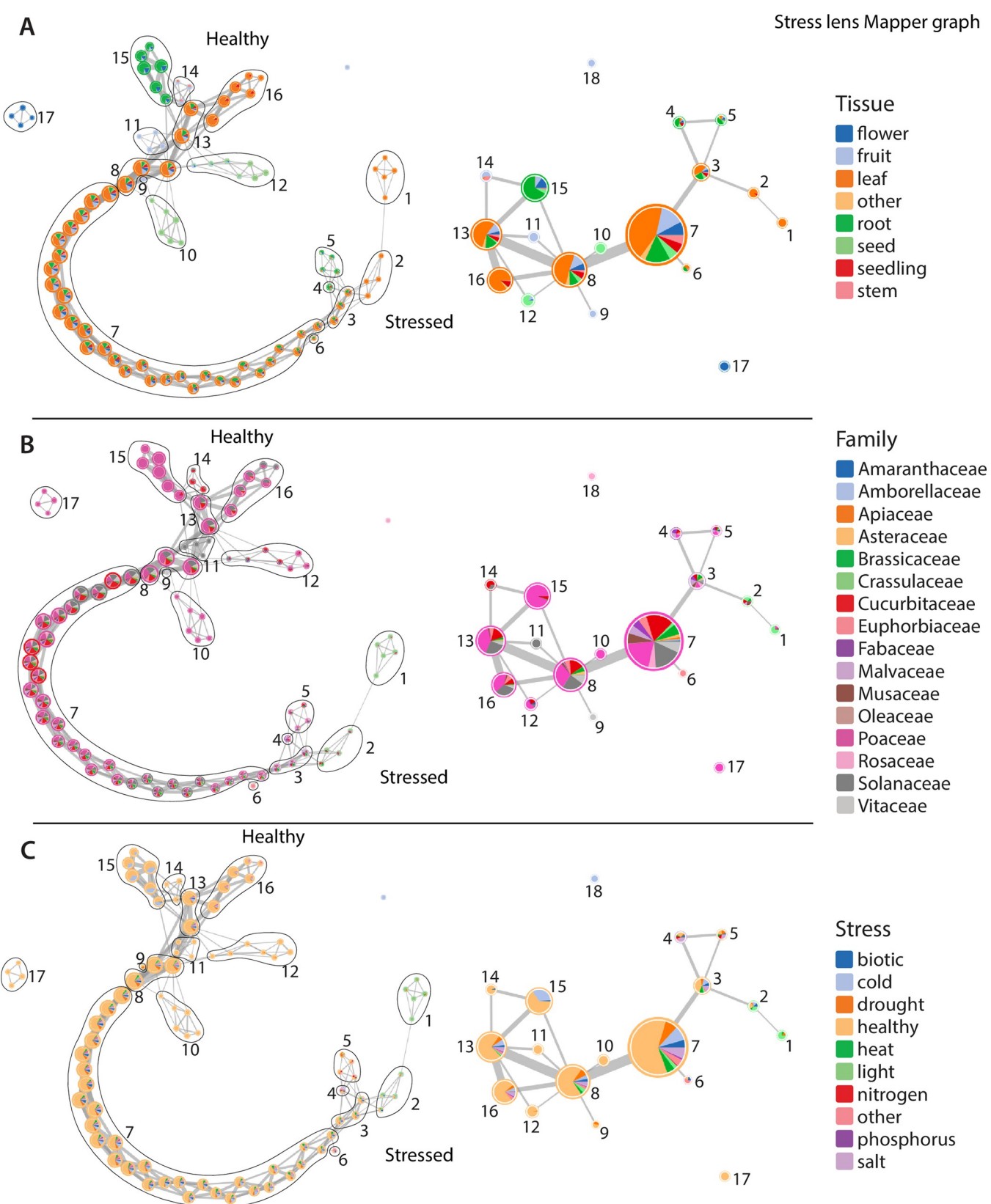

**Fig 4. Simplified Mapper graphs detailing the distribution of samples along the stress lens.** Nodes along the full Mapper graphs (left) are clustered into simplified Mapper graphs (right) and samples are colored by tissue (**A**), family (B), and stress category (C). Healthy and stressed ends of the Mapper graph are indicated.

but monocot flowers from the grass family (Poaceae) are found in discrete, branching nodes (9 and 17). The biotic and abiotic stress RNAseq samples are dispersed by tissue across the Mapper graph (Fig 3C), supporting the complexity and heterogeneity of these samples.

Mapper graphs clearly distinguish tissues across plant taxa, but what are the biological features that underlie this topology? We surveyed the expression patterns of the 6,328 orthogroups used to generate our Mapper graphs to see if they are enriched in certain biological processes related to evolutionarily conserved, tissue-specific functions. We classified genes as positively or negatively correlated with the tissue lens and conducted GO enrichment in these groups of genes. We expect negatively correlated genes to be characteristic of leaf gene expression and positively correlated genes to be characteristic of non-leaf gene expression. Supporting this, Mapper graphs and GO terms associated with the tissue lens–correlated genes point to photosynthetic versus nonphotosynthetic metabolism as a key factor in the overall gene expression patterns of plant tissues (Fig 3 and S1 Dataset). Enriched negatively correlated GO terms are mostly related to photosynthesis and include response to red and blue light, chloroplast and thylakoid organization, carotenoid metabolic process, and regulation of photosynthesis among others (S1 Dataset). Plants and green algae are characterized by a set of well-conserved genes that are not found in nonphotosynthetic organisms termed "the Green-Cut2 inventory" [16]. Most of the GreenCut2 genes (421 out of 677) are found within the 6,328 orthogroups in our analysis, and we tested if these are enriched among correlated genes. Genes from the GreenCut2 inventory are overrepresented in this set of genes, with 26.7% of the tissue-correlated (positively or negatively) genes being in the GreenCut2 resource versus 6.7% of the entire set of orthogroups (Table A in S1 Text). This overrepresentation is even more stark if we delimit our analysis to only the genes negatively correlated with the tissue lens, of which 50.3% are in the GreenCut2 inventory. The overlapping loci between the 2 sets contain genes encoding protein products involved in various aspects of photosynthesis, including pigment biosynthesis and binding (e.g., AT4G10340, AT1G04620, AT1G44446) [17–19], the operation of the photosynthetic light reactions (e.g., AT4G05180, AT5G44650, AT3G17930) [20–22], or the operation of the Calvin–Benson Cycle (AT1G32060) [23].

Enriched GO terms that are positively correlated with the tissue lens are largely related to housekeeping and core metabolic processes including ubiquitination, macromolecule catabolism, the electron transport chain, peptide biosynthesis, and Golgi vesicle–mediated transport among many others (S2 Dataset). Enriched genes include proteins involved in the TCA cycle and respiration (e.g., AT1G47420, AT2G18450, AT4G26910) [24–26] and in the development of specific nonphotosynthetic tissue types like seeds (e.g., AT2G40170, AT2G38560) [27,28] and pollen/pollen tubes (e.g., AT2G03120, AT2G41630) [29,30]. However, many of the tissue lens–correlated genes do not intuitively relate to the photosynthetic versus nonphotosynthetic tissue distinction, and further examination of these loci on a gene-by-gene basis may shed light on conserved differences between plant tissues.

The simplified Mapper graph from the stress lens has 18 nodes that form a continuous gradation of healthy to stressed tissues (Fig 4). Individual tissue types, regardless of stress condition, are enriched in certain nodes but are less defined than under the tissue lens (Fig 4A). RNAseq samples related to light and heat stress are found in discrete nodes (1 and 2, respectively) at the terminus of the Mapper graph across all species where these data were available (Fig 4C). Other stress RNAseq samples are found in nodes with healthy tissues but are

generally concentrated toward the stress end of the Mapper graph. An interesting exception is a group of cold stressed root samples from the grass (Poaceae) family (node 15). Clustering of distinct stresses within the same node suggests a core stress response conserved across Angiosperms for all abiotic and biotic factors. The gradient of sample distribution from healthy to stressed across the Mapper graph may be related to the severity of stress experienced by plants in each individual experiment.

To explore what constitutes these conserved stress-related expression patterns, we searched for GO enrichment of genes that are positively correlated with the stress lens. This group of genes is heavily enriched in functions related to stress, including responses to water deprivation, chitin, reactive oxygen species, fungi, wounding, bacteria, and general defense mechanisms (S3 Dataset). Genes positively correlated with the stress lens include loci related to the biosynthesis of compounds with diverse stress-related activities like jasmonic acid and jasmonic acid derivatives (AT2G35690, AT2G46370) [31,32] and ascorbic acid (AT3G09940) [33]. Negatively correlated genes are enriched in functions related to growth and reproduction such as DNA replication, mitosis, and rRNA processing, among others (S4 Dataset). This includes genes involved in regulation of the cell cycle (AT3G54650, AT4G12620, AT2G01120) [34–36], chromatin organization (AT1G15660, AT1G65470) [37,38], and the development of reproductive structures (AT1G34350, AT2G41670, AT4G27640, AT3G52940) [39–42]. This pattern points towards an intuitive distinction between the stressed and unstressed samples in our dataset in terms of their investment in cell proliferation and reproduction. Most of these genes are involved in core biological functions with conserved roles across eukaryotes, and their coordinated perturbation could be predictive of stress responses in diverse lineages.

## Discussion

Genome-scale datasets have high dimensionality, and even the simplest pairwise experiment has hundreds or thousands of complex and interconnected cellular pathways in dynamic flux between conditions. Comparisons across plant lineages are similarly complex, as each species has its own evolutionary history with thousands of duplicated, lost, or new genes enabling its unique and elegant biology. This complexity presents major challenges for characterizing underlying biological mechanisms and identifying shared and distinct properties across evolutionary timescales. Here, we leveraged the wealth of public gene expression datasets across diverse flowering plants and used a set of deeply conserved genes to search for patterns of conservation across tissue types, stress responses, and evolution. We first tested traditional dimensionality reduction and clustering-based approaches but found that they were largely ineffective and unable to clearly resolve samples. Instead, we used a novel topological framework to compare samples and test for evolutionary conservation.

Topological data analysis has been applied to complex, high dimensionality biological datasets including gene expression profiles correlated with human cancers and other diseases [5,43,44]. To our knowledge, TDA has not been used for plant science datasets outside of shape [45–47]. Flowering plants have tremendous phylogenetic, developmental, phenotypic, and genomic scale diversity, creating additional layers of complexity compared to other lineages. Despite this, Mapper was able to capture hidden and emergent signatures of gene expression at the tissue and stress scales that were missed using traditional approaches. Most developmental tissues or stress responses are not perfectly separated but instead fall within a gradient along a central shape. The central shape of the tissue lens Mapper graph represents the life cycle of a plant with transitions from the vegetative tissues of leaves and roots to reproductive flowers, fruit, and, eventually, seeds. Nodes along the Mapper graphs that contain mixtures of tissues such as fruits and flowers, leaves and stems, or even leaves and roots reflect

developmental plasticity, heterogeneity, and overlapping functions between different organs. Flowers give rise to fruits and the complex processes of fertilization, seed, and fruit development blur the lines between distinct tissue types. This complexity and interconnectivity is central to biological processes but is masked by traditional dimensionality reduction approaches, which can oversimplify nonlinear datasets.

The stressed and healthy samples are less clearly delineated in the Mapper graphs than samples from different plant tissues. This may reflect artifacts stemming from variation in the severity, duration, or method of applying stresses across different experiments and species. For example, mildly stressed samples might have expression signatures that mirror healthy tissues with comparatively few differentially expressed genes. Despite this issue, we observed a strong gradient of sample distribution from healthy to stressed across the graph. Distinct stresses were generally found within the same nodes, and genes that were positively correlated with the stress lens show enrichment in classical stress pathways. This includes the core stress-responsive hormones jasmonic acid and abscisic acid and their corresponding transcriptional network as well as broader shifts in metabolic processes geared toward defense. Taken together, this suggests that plants have deeply conserved expression signatures across evolution and for different stresses. Abiotic and biotic stress responses have been mostly studied in isolation, but they typically co-occur in natural environments, and they have overlapping signaling, hormonal, and network responses in plants (reviewed in [48]). The topological shape of gene expression points to a shared set of pathways or perturbations that define if a tissue is healthy or stressed. Environmental stresses broadly disrupt photosynthesis and core metabolic and cellular functions either as a direct response to physical trauma or in preparation for defense or resilience. These changes may serve as the backbone of the topological shape we observed for the stress lens.

Although we observed a deeply conserved pattern of gene expression underlying plant form and function, our analyses capture a snapshot of the evolutionary innovations found in flowering plants. We used a set of low-copy, conserved genes to enable comparisons of expression across species, and we had to exclude around approximately 70% of all plant genes. This includes most enzymes, transcription factors, and regulatory elements, which are mostly found in large, rapidly evolving, or lineage-specific gene families that cannot be resolved to high-confidence orthologs across eudicots and monocots. Duplication and subsequent sub- or neofunctionalization of these genes drive the evolution of new plant traits and developmental differences of plant organs. Single-copy genes by contrast have deeply conserved functions in core metabolism, photosynthesis, and housekeeping processes that typically transcend tissue, species, and environmental changes. Given these limitations, it is somewhat surprising that our analyses were able to clearly separate tissue types and stresses despite missing information from most of the genes that should underlie these biological differences. Applying TDA with a full set of genes in a single species with well-curated gene expression profiles could uncover complex or emergent biological signatures that were previously hidden.

Here, we provide a proof of concept for studying complex biological traits using TDA, and a similar analytical framework could be applied to numerous areas of plant science research and beyond. Compared to the approximately 300,000 published plant gene expression datasets [1], our study has a somewhat sparse sampling of species and a subset of expressed genes, yet we were able to detect a number of hidden trends. TDA of high-resolution sampling over narrower phenotypic spaces such as drought responses in a single species or tissue divergence across 900 million years of plant evolution could yield transformative insights that were previously overlooked. However, researchers should exercise caution when applying TDA to gene expression data as the lack of a robust hyperparameter tuning procedure could potentially result in misleading conclusions. This reflects a broader problem in machine learning and data

science, but hyperparameter search, cross-validation, and feature selection can enable data-driven tuning of the appropriate hyperparameters. With the appropriate datasets and sufficient sampling, TDA can be widely applicable for developing a deeper understanding of complex, emergent biological phenomena.

## Methods

### Assembling a representative catalog of flowering plant expression data

We selected species that captured the broadest phylogenetic diversity within angiosperms and species that had a breadth of expression at the tissue and stress levels. We also selected only species with a high-quality reference genome to enable accurate read mapping and down-stream comparative genomics. Metadata including species, accession, tissue type, experimental treatments, replicate number, and sequencing platform were collected manually for each sample using the NCBI BioProject and SRAs, as well as the primary data publications (S6 Dataset). Raw RNAseq reads were downloaded from the NCBI SRA and quantified using a pipeline developed in the VanBuren lab to trim, quantify, and identify differentially expressed genes (https://github.com/pardojer23/RNAseqV2). Using a common analytical pipeline helped reduce noise between experiments that used different algorithms in the original publications. Raw Illumina reads from various platforms were first quality trimmed using fastp (v0.23) [49] with default parameters. The quality filtered reads were pseudoaligned to the corresponding transcripts (gene models) for each species using Salmon (v1.6.0) [50] with the quasi-mapping mode. Transcript-level estimates were converted to gene-level transcript per million counts using the R package tximport [51].

### Comparing expression across species

To facilitate detailed cross-species comparisons, we first clustered proteins from all 54 species into orthogroups using Orthofinder (v2.3.8) [10]. Genomes and proteomes were downloaded for each species from Phytozome v13 [52]. Orthofinder was run using default parameters and the reciprocal DIAMOND search (v2.0.11) [53] was used for sequence alignment, and groups of similar proteins were clustered using the Markov Cluster Algorithm. In total, 2,317,289 genes (94% of input genes) were clustered into 86,185 orthogroups across the 54 species. Of these, 33,585 orthogroups are found in only a single species and 7,742 are found in at least 52 out of 54 species. This set of broadly conserved orthogroups was further refined by filtering out orthogroups with an average of >2 genes per ortholog for the diploid species to avoid including multigene families with diverse functions in the analysis. This set of 6,335 orthogroups was used as a common framework to allow comparison of expression across species. For orthogroups where a species had more than one gene, the total TPM for all genes in that orthogroup was summed and the raw TPM was used for single-copy genes. Expression data for each sample across all species were combined into a single expression matrix (S7 Dataset), and SVA was used to characterize the potential impacts of unmodeled technical variables on the dataset (see Text A in S1 Text). PCA was performed using built-in functions in Scikit-learn [54] on the log2+1 or z-score transformed gene expression data (raw TPMs) to reduce dimensionality and capture the main sources of variation within the datasets.

### Surrogate variable analysis

To account for batch effects and the influence of unmodeled factors on the expression matrix used for the present study, we applied SVA to generate estimates of surrogate variables and their effects on our expression matrices [55,56]. Briefly, SVA assumes that the expression of a

particular gene $i$ across $j$ independent RNA-seq experiments can be described by the following linear equation:

$$x_{ij} = u_i + f_i(y_j) + e_{ij} \tag{1}$$

where $u_i$ is the baseline expression level of gene $i$, $f_i(y_j)$ represents the effect of a measured variable $y_j$, and $e_{ij}$ is the error term [55]. However, if there are a number of $L$ unmodeled factors affecting the expression of gene $i$, then the error term $e_{ij}$ contains both randomly distributed experimental error as well as the effects of unmodeled factors. That is:

$$e_{ij} = \Sigma_l^L y_{li} g_{ij} + e\prime_{ij} \tag{2}$$

where $g_l = (g_l = (g_{l1}, \ldots, g_{ln})$ is a function describing the effect of all unmodeled factors up to $L$, $y_{li}$ is the coefficient describing the influence of an unmodeled factor $l$ on the expression of gene $i$, and $e\prime_{ij}$ is the true randomly distributed noise term [55]. Combining (1) and (2) yields:

$$x_{ij} = u_i + f_i(y_i) + \Sigma_l^L y_{li} g_{ij} + e\prime_{ij} \tag{3}$$

By using the svaseq() method implemented in the R package sva (v. 3.36.0) [56,57], we identified and estimated the values of 24 separate surrogate variables. These surrogate variables, which correspond to vectors of values for each expression value $x_{ij}$, in the $\Sigma_l^L y_{li} g_{ij} + e\prime_{ij}$ term in (3).

To determine the amount of variation due to a proxy batch variable (bioproject), 3 biological primary variables (stress, tissue, and family), and the pairwise interactions each surrogate variable explains, we regressed all the estimated surrogate variables on each variable (either batch or biological) or on a pairwise interaction. McNemar's formula was used to calculate the adjusted R2 values for each surrogate variable.

## Mathematical basis of topological data analysis

The flexibility of Mapper allows us to apply it to various types of data. Here, we will describe the Mapper construction in the simplest setting of point cloud data and then explain how it was applied to the gene expression data.

Consider a point cloud $X \subset \mathbf{R}^d$ equipped with a function $f: X \rightarrow \mathbf{R}$. An open cover of $X$ is a collection $U = \{U_i\}_{i \in \mathbf{I}}$ of open sets in $\mathbf{R}^d$, such that $X \subset \bigcup_{i \in \mathbf{I}} U_i$, where $\mathbf{I}$ is an index set. The 1-dimensional nerve of the cover $U$, denoted as $M := N_1(U)$, is called the Mapper graph of $(X, f)$. In this graph, each open set $U_i$ is represented as a vertex $i$, and 2 vertices, $i$ and $j$, are connected by an edge if and only if the intersection of $U_i$ and $U_j$ is nonempty.

To construct a Mapper graph, we start by defining a cover $V = \{V_j\}_{j \in \mathbf{J}}$ of the image $f(X) \subset \mathbf{R}$ of $f$, where $\mathbf{J}$ is a finite index set, by splitting the range of $f(X)$ into a collection of overlapping intervals. Next, for each $V_j$, we identify the subset of points $X_j$ in $X$ such that $f(X_j) \subset V_j$ and apply a clustering algorithm to identify clusters of points in $X_j$. The cover $U$ of $X$ is the collection of such clusters induced by $f^{-1}(V_j)$ for each j. Once we have the cover $U$, we compute its 1-dimensional nerve $M$ and visualize it in the form of a weighted graph.

For example, consider Fig 2A–2E. The point cloud $X$ in this case consists of points in the 2-dimensional plane, in the shape of a "Y". The function $f$ simply maps each point to its $y$-coordinate. We divide the range of $f$ into 4 overlapping intervals, represented by the 4 colored segments along the $y$-axis in Fig 2. For each interval $V_j$, the colored rectangles in the center panel of the figure show the subsets of points $X_j \in X$ such that $X_j = f^{-1}(V_j)$. Then, we apply clustering to each $X_j$ separately to obtain the cover $U$ of $X$. The 1-dimensional nerve of $U$, i.e., the mapper graph $M$, is shown in the rightmost panel. The color of each vertex corresponds to the

cover interval it belongs to. Fig 2A–2E illustrates mapper graph construction from the same set of points, but with *x*-coordinate used as the lens. We can observe that the 2 lens functions produce 2 slightly different mapper graphs.

### Constructing Mapper graphs and lens functions

To construct Mapper graphs from our gene expression data, we create 2 different lenses, adopting an approach similar to the one used in Nicolau and colleagues' paper. We refer to these lenses as the tissue lens and the stress lens, respectively. To create the stress lens, we first identified all the healthy samples from the dataset and fit a linear model to them. This model serves as the idealized healthy orthogroup expression. Then, we project all the samples (healthy as well as stressed) onto this linear model and obtain the residuals. These residuals measure the deviation of the sample gene expression from the modeled healthy expression. The lens function is simply the length of the residual vector. To define the cover, we divide the range of the lens function into intervals of uniform length, with the same amount of overlap between adjacent intervals. We experimented with a range of values length of the intervals and the size of the overlap to identify the values that produced relatively stable Mapper graphs. The clustering was performed using DBSCAN, a commonly used clustering algorithm for Mapper.

The construction of Mapper graph relies on several user-defined parameters: the lens function *f*, the cover $V$, and the clustering algorithm. Optimizing these parameters is an interesting open problem in TDA research [58]. The function *f* plays the role of a lens, through which we look at the data, and different lenses provide different insights [4]. The choice of *f* is typically driven by the domain knowledge and the data under consideration. In this study, the data under consideration are very similar to the dataset studied by Nicolau and colleagues [5]. Therefore, we followed similar methods to define the lenses. Our choice of lenses is further justified by the observations from the dimension reduction plots.

The cover $V = \{V_j\}_{j \in J}$ of $f(X)$ consists of a finite number of open intervals as cover elements. To define $V$, we use the simple strategy of defining intervals of uniform length and overlap. Adjusting the interval length and the overlap increases or decreases the amount of aggregation provided by the Mapper graph. The optimal choice was made by visually inspecting Mapper graphs over a range of parameter values. The parameters resulting in the most stable structure were selected. Any clustering algorithm can be employed to obtain the cover $U$. We use the density-based clustering algorithm, DBSCAN [59], which is commonly used in Mapper because it does not require a priori knowledge of the number of clusters. Instead, DBSCAN requires 2 input parameters: the number of samples in a neighborhood for a point to be considered as a core point, and the maximum distance between 2 samples for one to be considered in the neighborhood of the other.

### Functional annotation of orthogroups

The correlation between expression values and tissue lens and stress lens values was calculated for each orthogroup. The top 2.5% most positively and negatively correlated orthogroups for each lens were selected to represent the tissue lens or stress lens correlated orthogroups. Arabidopsis gene IDs were used to identify the overlap between the GreenCut2 [16] inventory with Arabidopsis orthologs in our overall set of orthogroups, as well as our sets of tissue lens and stress lens correlated orthogroups. The *binom_test()* function from SciPy [60] was used to apply one-sided binomial tests to check for enrichment of GreenCut2 loci in the overall, tissue lens, and stress lens correlated orthogroup sets. GO term enrichment of the sets of genes mapped to orthogroups and correlated with the tissue lens or stress lens was done using GOA-TOOLS [61]. Data on gene function and biochemical reactions associated with specific loci

were derived from TAIR [62], KEGG [63], and a genome-scale metabolic model of Arabidopsis metabolism from [64].

## Supporting information

**S1 Text. Fig A. Histogram of 3-way factors of the RNAseq samples before and after downsampling**. The distribution of 3-way factors for family, tissue, and stress is plotted. The 16 families, 8 tissue types, and 10 stresses equate to 1,280 unique 3-way combinations, but we only observed 195 unique combinations in our dataset. The distribution of samples from the entire dataset is shown on the left, and the distribution of samples when downsampling the 30 most common 3-way combinations is shown on the right. Raw expression data underlying the graphs in this figure can be found in S7 Dataset, and code can be found in https://zenodo.org/records/8428609 [65]. **Fig B. Factor-wise frequency plots of RNAseq samples before and after subsampling.** The number of samples in each family, tissue type, or stress is plotted before (top) and after (bottom) subsampling. Raw expression data underlying the graphs in this figure can be found in S7 Dataset, and code can be found in https://zenodo.org/records/8428609 [65]. **Fig C. Topology of Mapper graphs generated from the subsampled data.** Samples from each node in the Mapper graph are colored by plant family (A), stress (B), or tissue type (C), using the subsampled data. The overall topology and sample distribution are similar to the Mapper graphs constructed with the full, unbalanced dataset, suggesting that sample distribution is not a major factor in our analyses. **Fig D. Linear regression analysis of association of surrogate variables to one batch variable (BioProject), our biological variables of interest (stress, tissue, and family), and their pairwise interactions.** All surrogate variables were regressed on either each variable or interaction individually to calculate adjusted $R^2$ values. **Table A. Enrichment of GreenCut2 genes in orthogroup-mapped *Arabidopsis thaliana* genes and stress-/tissue-correlated orthogroup-mapped genes.** The proportion of GreenCut2 genes in the all the orthogroups used in this study was compared against the proportion of GreenCut2 genes in a list of all *A. thaliana* genes using a one-sided binomial test. The proportion of tissue lens and stress lens correlated orthogroup-mapped genes in GreenCut2 was compared against the proportion of GreenCut2 genes in the entire set of orthogroup-mapped genes using one-sided binomial tests. Tissue-correlated genes were hypothesized to be more likely to be in GreenCut2 than a random selection of orthogroup-mapped genes, and the stress-correlated genes were hypothesized to be less likely.
(DOCX)

**S1 Dataset. GO term enrichment results on genes negatively correlated with the tissue lens.**
(XLSX)

**S2 Dataset. GO term enrichment results on genes positively correlated with the tissue lens.**
(XLSX)

**S3 Dataset. GO term enrichment results on genes positively correlated with the stress lens.**
(XLSX)

**S4 Dataset. GO term enrichment results on genes positively correlated with the stress lens.**
(XLSX)

**S5 Dataset. Overlap between orthogroup-mapped genes and tissue lens and stress lens correlated genes with the GreenCut2 resource (Karpowicz).**
(XLSX)

**S6 Dataset. Metadata of the raw data used in this experiment.**
(CSV)

**S7 Dataset. Expression matrix of TPMs for the normalized orthogroups.**
(CSV)

## Author Contributions

**Conceptualization:** Sourabh Palande, Joshua A. M. Kaste, Miles D. Roberts, Kenia Segura Abá, Jamell Dacon, Aman Y. Husbands, Arjun Krishnan, Beronda L Montgomery, Elizabeth Munch, Addie M. Thompson, Alejandra Rougon-Cardoso, Daniel H. Chitwood, Robert VanBuren.

**Data curation:** Sourabh Palande, Joshua A. M. Kaste, Miles D. Roberts, Kenia Segura Abá, Carly Claucherty, Jamell Dacon, Rei Doko, Thilani B. Jayakody, Hannah R. Jeffery, Nathan Kelly, Andriana Manousidaki, Hannah M. Parks, Emily M. Roggenkamp, Ally M. Schumacher, Jiaxin Yang, Sarah Percival, Jeremy Pardo, Alejandra Rougon-Cardoso, Daniel H. Chitwood, Robert VanBuren.

**Formal analysis:** Sourabh Palande, Joshua A. M. Kaste, Miles D. Roberts, Kenia Segura Abá, Carly Claucherty, Rei Doko, Thilani B. Jayakody, Hannah R. Jeffery, Nathan Kelly, Andriana Manousidaki, Hannah M. Parks, Emily M. Roggenkamp, Ally M. Schumacher, Jiaxin Yang, Sarah Percival, Jeremy Pardo, Alejandra Rougon-Cardoso, Daniel H. Chitwood, Robert VanBuren.

**Project administration:** Alejandra Rougon-Cardoso, Daniel H. Chitwood, Robert VanBuren.

**Software:** Jeremy Pardo.

**Supervision:** Daniel H. Chitwood, Robert VanBuren.

**Visualization:** Daniel H. Chitwood.

**Writing – original draft:** Joshua A. M. Kaste, Alejandra Rougon-Cardoso, Daniel H. Chitwood, Robert VanBuren.

**Writing – review & editing:** Aman Y. Husbands, Arjun Krishnan, Beronda L Montgomery, Elizabeth Munch, Addie M. Thompson, Alejandra Rougon-Cardoso, Daniel H. Chitwood, Robert VanBuren.

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
