## [Editor Report · Decision Letter 0]

31 Jan 2023

Dear Dr VanBuren, 

Thank you for submitting your manuscript entitled "The topological shape of gene expression across the evolution of flowering plants" for consideration as a Research Article by PLOS Biology. Thank you also for your patience as we completed our editorial process, and please accept my apologies for the delay in providing you with our decision.

Your manuscript has now been evaluated by the PLOS Biology editorial staff as well as by an academic editor with relevant expertise and I am writing to let you know that we would like to send your submission out for external peer review. However, I should note that the outcome of our discussion of your manuscript is that we have some reservations as to the the overall strength of novel biological insight offered by your data. We would need to be persuaded by the reviewers that the paper has the potential to offer the significant strength of advance that we require for publication in order to pursue it further for PLOS Biology.

Before we can send your manuscript to reviewers, we need you to complete your submission by providing the metadata that is required for full assessment. To this end, please login to Editorial Manager where you will find the paper in the 'Submissions Needing Revisions' folder on your homepage. Please click 'Revise Submission' from the Action Links and complete all additional questions in the submission questionnaire.

Once your full submission is complete, your paper will undergo a series of checks in preparation for peer review. After your manuscript has passed the checks it will be sent out for review. To provide the metadata for your submission, please Login to Editorial Manager (https://www.editorialmanager.com/pbiology) within two working days, i.e. by Feb 02 2023 11:59PM.

Kind regards,

Ines

--

Ines Alvarez-Garcia, PhD

Senior Editor

PLOS Biology

---

## [Decision Letter · Decision Letter 1]

5 Apr 2023

Dear Dr VanBuren,

Thank you for your patience while your manuscript entitled "The topological shape of gene expression across the evolution of flowering plants" was peer-reviewed at PLOS Biology. Please also accept my sincere apologies for the delay in providing you with our decision. The manuscript has now been evaluated by the PLOS Biology editors, an Academic Editor with relevant expertise, and by three independent reviewers. 

The reviews are attached below. As you will see, the reviewers find the method and conclusions of the manuscript novel and important for the field, but they also raise several concerns that would need to be addressed before we can consider the manuscript for publication. The reviewers think that you should state the limitations of the methods in more detail and address other methodological issues, along with other points that need to be clarified.

In light of the reviews and discussions with the Academic Editor and the rest of the team, we would like to invite you to revise the work to thoroughly address the reviewers' reports. You should address all the points related with the methodological aspect of the paper raised by Reviewer 1 and 2, but we don't think it is necessary to look at other embedding algorithms (UMAP), and present different PCs/tSNE dimensions, as suggested by Reviewer 2. In addition, you should change the article type to Methods and Resources when you submit the revision, as we do think the article would fit better that format.

Given the extent of revision needed, we cannot make a decision about publication until we have seen the revised manuscript and your response to the reviewers' comments. Your revised manuscript is likely to be sent for further evaluation by all or a subset of the reviewers.

**IMPORTANT - SUBMITTING YOUR REVISION**

3. Resubmission Checklist

a) *PLOS Data Policy*

b) *Published Peer Review*

c) *Blurb*

Please also provide a blurb which (if accepted) will be included in our weekly and monthly Electronic Table of Contents, sent out to readers of PLOS Biology, and may be used to promote your article in social media. The blurb should be about 30-40 words long and is subject to editorial changes. It should, without exaggeration, entice people to read your manuscript. It should not be redundant with the title and should not contain acronyms or abbreviations. For examples, view our author guidelines: https://journals.plos.org/plosbiology/s/revising-your-manuscript#loc-blurb

Sincerely,

Ines

--

Ines Alvarez-Garcia, PhD

Senior Editor

PLOS Biology

Reviewers' comments

Rev. 1:

In the manuscript entitled, "The topological shape of gene expression across the evolution of flowering plants", Palande et al. present an innovative new analysis method for identifying trends in large collections of data. They take transcriptomes of different tissues and under different stresses from species across the phylogeny of flowering plants and analyze them all together with topological data analysis. They represent the shape of gene expression across a tissue lens and find a continuum from leaves to seeds with some branches for specific tissues in various plant families. The genes underlying the difference in tissues generally relate to photosynthesis in leaves versus core metabolic processes such as ubiquitination and Golgi vesicle-mediated transport in non-leaf tissues. Their work on stress is less conclusive, as healthy and stressed samples are often mixed at nodes. However, they did identify a general trend from healthy to stressed. They also found a negative correlation between stressed tissues and growth/cell division, which is a well-known trade off plants make, suggesting fundamental biology can be accessed by these analysis methods. To my knowledge this is the first use to analyze expression patterns in plants, and there are only a handful of uses in all of biology. Overall, the method is novel and adds a key new tool to analyze complex high dimensional data, particularly across species.

Major concerns

1. We ask the authors to consider and further discuss possible limitations of their method in more detail. Specifically, how does the proportion of samples in the input data affect the weight of output terms? For example, the dataset analyzed has high numbers of samples with drought, heat, and salt stress, which may explain why water deprivation comes up as a major output term for stress response overall. Something else might come up if stress samples were more balanced. Likewise, because Poaceae are a large fraction of the sample species input, they might play a big part in shaping the stress outputs. It would be worth re-running the analysis without Poaceae to see what effect that has on the output. 

2. Similarly, the authors restrict the analysis to 6,328 orthologous low copy genes that are conserved across the angiosperm species used. This is the only way to do the analysis, but it eliminates a lot of important genes from the analysis. For instance, most of the transcription factors and signaling pathway components controlling development are members of large protein families that are excluded from this analysis. This undoubtedly tilts the tissue results toward more core biological functions such as photosynthesis, ubiquitination, endomembrane trafficking, etc., instead of classic developmental regulators. The authors should discuss how this limitation of genes included in the analysis shapes and limits the outputs. 

3. This paper is introducing topological data analysis to most readers for the first time, so it is imperative to create a better schematic diagram explaining the analysis method to the reader. Figure 2a and 2b are not sufficient. The text lines ~167-172 describes a step wise process to arrive at the final shape. It would be worth creating a detailed step by step diagram illustrating the analysis process that directly relates to the steps described. Also please label the lens function with what it is, not just lens function #1. 

Minor comments:

1. Line 54-55 "thousands of diverse plant species spanning over 900 million years of evolution" The sampled plants are all extant plants, so they do not span evolution. Maybe you mean some of the plants have diverged many million years ago? 

2. What is in the "Other" category for tissue and stress? 

3. Supplemental datasets 6 (mentioned in line 355) and 7 (mentioned in line 380) are missing from the submission (no link to them on the document). 

4. For Stress the authors say there are 17 nodes in text, but there are 18 nodes labeled on the figure. 

5. Please label which end is the stress end and which is the healthy end on Figure 4. 

6. In Figure 2F, what does the box mean? Why isn't there an equivalent box on the tissue side, in Figure 2E? 

7. Please check your color scheme for colorblind friendliness. Even for full color vision people, the pie charts in Figure 4 can be hard to see. 

8. Figure 3 seems to be mis-titled as stress lens, and is the tissue lens.

Rev. 2:

Palande et al report on a meta-analysis of transcriptomic data throughout the angiosperms. By simplifying highly dimensional gene expression data from many plant species with a topological simplification tool, Mapper, the authors have identified global features in the data reflecting developmental, taxonomic and environmental variables.

The approach is very original and the conceptual framework seems quite promising for synthesizing important features in the data. In principle, visualizations using dimensionality reduction can generate an ensemble of shapes, and as such interpreting these shapes could be misleading or even meaningless - I think this point should be emphasized in the introduction. How meaningful is the shape - what is the biological significance of having "loops, branch point, flares" (line 92)?

I think the manuscript should emphasize more clearly what is the advantage of presenting the results with mapper graphs, and what novel conclusions could we draw (the source/sink continuum is apparent on the PC plot). It may be useful to look at other embedding algorithms (UMAP), and present different PCs/tSNE dimensions. 

How biased is the dataset to species or stresses? It appears less biased to tissues, where most of the variation is detected. This is data with substantial sparsity and likely very sensitive to sampling.

More generally, I am not sure if gene expression having any "structure" or "shape" is the most appropriate way to phrase the findings of the study. The resulting "shape" is a representation of the gene expression, rather that a guiding force for resulting phenotypes (line 95). 

Line 120 - Amborella is not a "basal" angiosperm - please rephrase to "sister of the rest of the angiosperms (for in-depth discussion, please check Stacey Smith's blog)

Rev. 3:

This study by Palande et al. entitled The topological shape of gene expression across the evolution of flowering plants harnesses the data dimensionality reduction procedure Mapper to derive a topological understanding of gene expression across diverse plant species. The study explores 2,671 SRA samples from 54 diverse plant species which are phylogenetically joined by 6,328 conserved orthogroups. When exploring the differences between tissue expression and stress expression (abiotic vs biotic) the authors interpret their projection and modelling outcomes as distinct and conserved expression patterns across angiosperms that delineate different tissue types or responses to biotic and abiotic stresses based on their agreement with a particular lens-function (e.g. length of residual vector). 

While this study provides a promising opportunity to aggregate some of the vast sample space of the NCBI SRA database, it fails to deliver substantiating evidence and presents significant analytical, bioinformatic, and conceptual shortcomings that would need to be addressed in a major revision. If their conclusions still hold true after the sufficient scrutiny of (yet missing) negative controls and cross-sample data normalization, this study could provide value for a specialised audience that seeks to replicate their topological procedure for other SRA samples. 

Major comments:

Overall, the two most important resources for reproducibility and study assessment were not provided for review by the authors: The final expression dataset (Suppl Data 7) and the GitHub repository (Data availability: The code, metadata, and raw datasets from this project are available on a dedicated GitHub page: <empty space>).

Methodological shortcomings:

- Assembly of gene expression data: While the intuition to generate TPM values based on a standardized pipeline is correct, the lack of any normalization across either tissues or across species or across stress conditions or across similar data quality is making the procedure of deriving comparable expression level estimates highly biased and insufficient. Without sufficient normalization and accounting for sequencing depth, coverage, sequencing technology conformity, contamination checks, batch effect correction, etc any observed topological pattern could be explained by these technical artifacts. Since none of these quality-control steps were performed and no controls to suggest otherwise were presented, I am not convinced that the presented topologies represent true biological signature. I have to assume that these patterns are largely driven by differences in their technical treatment and highly diverse experimental designs.

- Mathematical basis of topological data analysis: The authors present the Mapper approach as the only alternative compared to established dimensionality reduction methods that was able to present them with patterns they felt comfortable to interpret (p. 8 "We first tested traditional dimensionality reduction and clustering-based approaches but found they were largely ineffective and unable to clearly resolve samples. Instead, we used a novel topological framework to compare samples and test for evolutionary conservation."). How can the authors ensure that traditional dimensionality reduction and clustering-based approaches didn't fail, because they (in fact intrinsically) captured the bias of the non-standardized data? How would the authors argue against the argument that they cherry-picked a method that allowed them to show any pattern? The Mapper approach is also known to have analogous weaknesses as traditional dimensionality reduction methods such as determining an appropriate and agnostic number of overlapping intervals used to cover the data and the robustness of diverse clustering algorithm (and distance metric) able to consistently group similar points together. I interpret the claim "We experimented with a range of value lengths of the intervals and the size of the overlap to identify the values that produced relatively stable mapper graphs." as visually/manually cherry-picking the most favourable topologies rather than relying on a robust and objective metric. 

The fact that furthermore "The clustering was performed using DBSCAN, a commonly used clustering algorithm in Mapper [14]." without demonstrating the robustness of topologies to differences in clustering methods further strengthens my suspicion. The claim that other studies also used the Mapper approach on gene expression data despite the open problem in TDA research to optimize Mapper parameters is not convincing at this stage where traditional dimensionality reduction methods fail.

- For the claim that most of the community assumes that biotic and abiotic stress are evolutionarily independent (not clear to me what this means), no reference is presented and their finding that biotic and abiotic stress samples (therefore unexpectedly) showed similar topologies/patterns/distributions further illustrates the importance to standardize and normalize across SRA samples, tissues, species etc. The fact that biotic and abiotic stress samples show similar patterns/topologies may simply be the result of the fact that most stress responses involve a much smaller number of genes than tissue development or housekeeping and thus technical artifacts are more pronounced under these (tissue, housekeeping, non-stress) conditions. Since no control experiments are provided, I am not convinced that these stress similarities are in fact true biological signatures.

---

## [Decision Letter · Decision Letter 2]

16 Aug 2023

Dear Dr VanBuren,

Thank you for your patience while we considered your revised manuscript entitled "The topological shape of gene expression across the evolution of flowering plants" for publication as a Research Article at PLOS Biology. This revised version of your manuscript has been evaluated by the PLOS Biology editors, the Academic Editor and the original reviewers.

The reviews are attached below. Based on these comments, we are likely to accept this manuscript for publication, provided you address the remaining points raised by Reviewer 3. In addition, please address the data and other policy-related requests stated below.

In addition, we would like you to consider a suggestion to improve the title:

"Topological data analysis across the evolution of flowering plants reveals a core gene expression backbone that defines plant form and function"

We expect to receive your revised manuscript within two weeks. 

*Published Peer Review History*

*Press*

Sincerely,

Ines

--

Ines Alvarez-Garcia, PhD

Senior Editor

PLOS Biology

Fig. 1B-E; Fig. 2F, G; Fig. S1 and Fig. S2

**Please also provide at this stage all the accession numbers and DOIs.

BLURB

Please also provide a blurb which (if accepted) will be included in our weekly and monthly Electronic Table of Contents, sent out to readers of PLOS Biology, and may be used to promote your article in social media. The blurb should be about 30-40 words long and is subject to editorial changes. It should, without exaggeration, entice people to read your manuscript. It should not be redundant with the title and should not contain acronyms or abbreviations. For examples, view our author guidelines: https://journals.plos.org/plosbiology/s/revising-your-manuscript#loc-blurb

Reviewers' comments

Rev. 1:

The authors have carefully and thoroughly addressed all of the points we have raised. Their further analysis has shown the robustness of their method to the imperfect distribution of public datasets. We commend the authors on this innovative work. 

Rev. 2:

Thank you for addressing my comments - the manuscript is ready for publication. 

Rev. 3:

I appreciate the authors efforts to address my concerns. Some of my concerns were successfully approached and are now resolved, but a few major concerns remain unresolved which need further attention. I therefore recommend to address these concerns in full in a minor revision.

Rev. 3 (previous comments):

This study by Palande et al. entitled The topological shape of gene expression across the evolution of flowering plants harnesses the data dimensionality reduction procedure Mapper to derive a topological understanding of gene expression across diverse plant species. The study explores 2,671 SRA samples from 54 diverse plant species which are phylogenetically joined by 6,328 conserved orthogroups. When exploring the differences between tissue expression and stress expression (abiotic vs biotic) the authors interpret their projection and modelling outcomes as distinct and conserved expression patterns across angiosperms that delineate different tissue types or responses to biotic and abiotic stresses based on their agreement with a particular lens-function (e.g. length of residual vector). 

While this study provides a promising opportunity to aggregate some of the vast sample space of the NCBI SRA database, it fails to deliver substantiating evidence and presents significant analytical, bioinformatic, and conceptual shortcomings that would need to be addressed in a major revision. If their conclusions still hold true after the sufficient scrutiny of (yet missing) negative controls and cross-sample data normalization, this study could provide value for a specialised audience that seeks to replicate their topological procedure for other SRA samples. 

Major comments: 

Overall, the two most important resources for reproducibility and study assessment were not provided for review by the authors: The final expression dataset (Suppl Data 7) and the GitHub repository (Data availability: The code, metadata, and raw datasets from this project are available on a dedicated GitHub page: <empty space>). 

Authors: We apologize for the oversight here. Supplemental Dataset 7 is now available on GitHub and in the revision. This file is relatively large (~200 Mb), and we were not able to upload it in the first submission. The GitHub link was corrected, and is now active as well: https://github.com/PlantsAndPython/plant-evo-mapper

Methodological shortcomings: 

Authors: Assembly of gene expression data: While the intuition to generate TPM values based on a standardized pipeline is correct, the lack of any normalization across either tissues or across species or across stress conditions or across similar data quality is making the procedure of deriving comparable expression level estimates highly biased and insufficient. Without sufficient normalization and accounting for sequencing depth, coverage, sequencing technology conformity, contamination checks, batch effect correction, etc any observed topological pattern could be explained by these technical artifacts. Since none of these quality-control steps were performed and no controls to suggest otherwise were presented, I am not convinced that the presented topologies represent true biological signature. I have to assume that these patterns are largely driven by differences in their technical treatment and highly diverse experimental designs 

This is an important point and we agree that standardization across the experiment is essential. We feel our analyses are highly standardized, statistically robust, and largely free of technical artifacts beyond the inherent noise of RNAseq data. 

Rev-3: While I appreciate the authors' confidence in the SRA database, would it be possible to specifically state how normalization ACROSS diverse SRA samples were performed to provide concrete evidence for their strong claim that "[…] our analyses are highly standardized, statistically robust, and largely free of technical artifacts beyond the inherent noise of RNAseq data"? Did the authors rely on the "SRA Normalized Format" (default, see e.g. https://www.ncbi.nlm.nih.gov/sra/docs/data-format-faq/) as input to their analyses? If yes, did the authors check whether the samples giving strongest biological signature (especially the ones not captured by alternative dimensionality reduction approaches) still have a high "original quality score" in the non-SRA normalized samples (see FAQs in the previous link)? This important analysis detail should be made clear in the manuscript.

Authors: Below we include a detailed summary of the various QC metrics we used and clarified this in text (including a new supplemental note about surrogate variable analysis). 

We used Surrogate Variable Analysis (SVA) (Leek et al. 2012) to explore the effects of confounding technical variables on the publicly available SRA data assembled for this study. Briefly, we identified three primary variables of interest (tissue, stress, and family), which were fixed in the model used to estimate "surrogate variables" to minimize the amount of variability attributable to these primary variables captured by the estimated surrogate variables (see Supplementary Methods for Surrogate Variable Analysis). These surrogate variables represent unaccounted for technical variables impacting the dataset. Due to the breadth of families, stresses, and tissues analyzed, we do not have a full factorial design (i.e., there are combinations of family, stress, and tissue factor values for which there are no expression datasets). Because of this, SVA would remove variability due to our primary variables and their interactions. To get a sense of what kind of impact the surrogate variables might have on the dataset when removed, we estimated the correlation between the first order interactions between our primary variables and the surrogate variables identified by SVA. We identified 24 surrogate variables which individually captured between 53% and 98% of variation between BioProjects (Supplemental Figure 4). We also estimated the interaction terms between the tissue, family, and stress factor combinations that were present in the dataset and estimated how much of their variation was getting captured by the surrogate variables. Individual surrogate variables captured up to 14% of variation between stress conditions, up to 66% of variation between tissue conditions, and up to 63% of variation between families. For the interaction terms between primary variables, individual surrogate variables captured up to 83% of the variation between tissue and family combinations, up to 65% of the variation between stress and family combinations, and up to 71% of the variation between tissue and stress combinations. This suggests that even though stress, tissue, and family are treated as protected primary variables, there are underlying latent variables related to our primary variables and their interactions that may be important sources of biological variation being captured by the surrogate variables. Although individual surrogate variables could be selectively accounted for in downstream analyses in such a way that minimizes the removal of biological signal, this would be a highly subjective process. Moreover, due to our inability to precisely calculate the true correlation between our surrogate variables and interaction terms due to the fact that many factor combinations are missing, this would be statistically dubious as well.

Because the surrogate variables show substantial linear correlation with our primary variables and their interaction terms, the application of SVA would require eliminating substantial amounts of biological signal. Since the goal of our study is to identify heterogeneous patterns due to stress, tissue, and family within a high-dimensional gene expression dataset, SVA may not be appropriate for us to use. Alternatively, one could potentially minimize the loss of this signal by cherry-picking individual surrogate variables to include in downstream analysis, which would naturally introduce human bias. A third option would be to use an algorithm like ComBat-seq (Zhang, Parmigiani, and Johnson 2020) that relies on explicitly defined batches, which is problematic for the present study since the closest metadata for batch available for the studies gathered on SRA is the BioProject ID's, but these are, at best, a proxy for batches of samples and are not sufficient to assess the technical variability or noise in the data. More broadly, as discussed in (Jaffe et al. 2015), such genomic data "cleaning" methods, by their very nature, delimit the observable features of the resulting datasets to those prespecified by the investigator. In our view, this limits their utility for broad exploratory analyses of the kind described in this study. For all the above reasons, we opted to not use SVA, ComBat, or related techniques prior to downstream analyses. These shortcomings also emphasize the need for tools like Mapper that can, as shown in this manuscript, reveal patterns that are amenable to downstream analysis. 

Rev-3: Thank you for this detailed assessment of why their input samples are too heterogeneous or too sparsely sampled for sufficient batch effect correction. This was actually one of my major concerns of this study that if classic batch-effect smoothing/removal methods fail, dimensionality reduction will only project this shortcoming into a lower dimensional space. The authors' argument that since their input data represents an insufficient sampling to enable various batch effect removal methods, their dimensionality reduction approach can reveal true biological patterns is not clear to me, since no data or evidence is presented for this claim. The core principle of batch effect removal relies on exploring the nature of the variance in sufficiently sampled data and so does dimensionality reduction. Thus, both methods should be sensitive to insufficient sampling and data heterogeneity. Do the authors disagree? 

Authors: The raw TPM values have some degree of standardization by library size (i.e., number of reads per sample), and we transformed all expression values by Z-score prior to any downstream analyses. 

Rev-3: Thank you for clarifying. Was this Z-score standardization performed on the "SRA Normalized Format" (default, see e.g. https://www.ncbi.nlm.nih.gov/sra/docs/data-format-faq/) or on the SRA raw samples (non-SRA normalized samples)? This important analysis detail should be made clear in the manuscript.

Authors: The Z-score enables cross-species comparisons as values within each dataset are normalized to a common scale based on their standard deviation and mean rather than absolute values. Dimensionality reduction (t-SNE and PCA) from the z-score transformed expression in Figure 1 shows a clear separation of different plant tissues across species, suggesting we are identifying real developmental patterns in our dataset and not technological artifacts. Labeling the samples by technology, year the dataset was published, BioProject, or other variables showed no correlation. 

Rev-3: Excellent! Would it be possible to add this analysis ("Labeling the samples by technology, year the dataset was published, BioProject, or other variables showed no correlation.") as Supplementary Figures? This should give readers more confidence in the methodology. 

Authors: Tissue patterns are quite clear, and there is no reasonable way that technical artifacts could be causing this delineation as any artifacts or variability would be found within all sample types and species and not within a specific factor such as tissue, stress, or species that could create a misleading pattern. Furthermore, GO term analysis identified sets of genes that are consistent with the lens function we were using, such as photosynthetic genes delineating leaves from other tissues and stress responsive genes delineating healthy from unhealthy tissues. If our analyses were picking up on technical artifacts instead of biological patterns, there should be no enriched GO terms that are consistent with our classification. 

Rev-3: Thank you for performing this quality control analysis. I agree with the authors that this is promising evidence. Did the authors have a chance to confirm this analysis with a negative control whereby samples with clear signatures of technical artifacts do not pick up enriched GO-terms that would be consistent with their lens function? If this is indeed the case, then readers will appreciate this analysis by placing more confidence in their method.

Authors: We hope this clarifies the concerns raised by this reviewer and showcases the utility of our dataset and approach. 

- Mathematical basis of topological data analysis: The authors present the Mapper approach as the only alternative compared to established dimensionality reduction methods that was able to present them with patterns they felt comfortable to interpret (p. 8 "We first tested traditional dimensionality reduction and clustering-based approaches but found they were largely ineffective and unable to clearly resolve samples. Instead, we used a novel topological framework to compare samples and test for evolutionary conservation."). How can the authors ensure that traditional dimensionality reduction and clustering-based approaches didn't fail, because they (in fact intrinsically) captured the bias of the non-standardized data? How would the authors argue against the argument that they cherry-picked a method that allowed them to show any pattern? The Mapper approach is also known to have analogous weaknesses as traditional dimensionality reduction methods such as determining an appropriate and agnostic number of overlapping intervals used to cover the data and the robustness of diverse clustering algorithm (and distance metric) able to consistently group similar points together. I interpret the claim "We experimented with a range of value lengths of the intervals and the size of the overlap to identify the values that produced relatively stable mapper graphs." as visually/manually cherry-picking the most favourable topologies rather than relying on a robust and objective metric. 

Authors: We are receptive to the concern that, due to the lack of concordance between dimensionality approaches like PCA and our Mapper graphs and the lack of a robust hyperparameter tuning procedure for TDA, it may appear that we are cherry-picking favorable topologies to present.

Rev-3: Thank you for confirming that TDA lacks a robust hyperparameter tuning procedure. My major concern in this context was not that the authors cherry-picked their results, but that users of their method when broadly employed to various datasets may feel inclined to cherry-pick in the absence of an unbiased hyperparameter tuning procedure. This point needs to be extensively discussed in the main manuscript and clear guidelines presented to avoid (un)intentional cherry-picking by the end user.

Authors: We believe it is reasonable to assume that if the structures we are presenting are either (a) highly sensitive to the exact set of samples in the dataset, or (b) not reproducible using intuitive related, but distinct, lens functions, this would undermine the strength of our results. However, as shown in our responses demonstrating robustness to downsampling and the supplemental data showing results from using roots as a lens, our results are robust in both cases. This gives us confidence in the results we are presenting. 

Rev-3: While I agree with this statement and appreciate the analysis, the concern lies rather with the fact that less nuanced input data (as could be explored by users of the TDA method) could demonstrate less robustness (which was never tested by the end user). Thus, this type of "robustness confirmation analysis" should be clearly communicated to the user.

Authors: We tested the most commonly used dimensionality reduction approaches for expression-based datasets including MDS, t-sne, and PCA as well as hierarchical clustering approaches, and it is certainly possible that other methodologies could produce similar topologies to our TDA based analyses, but we are quite skeptical. We agree with the reviewer that these analyses may be picking up technical differences between experiments such as variations in how, when, and which exact tissues were collected, differences in genotype, or variation in the duration or magnitude of stress. This is somewhat supported by our results from the surrogate variable analysis, which was unable to remove these artifacts without removing the variables we were testing (e.g, tissues, species, or stresses). 

Rev-3: I greatly appreciate that the authors confirm my major concerns and that established dimensionality reduction methods are not robust to technical or sampling artifacts.

Authors: This is really the core finding of our manuscript: TDA can find hidden biological structure in complex, noisy datasets that is missed by traditional dimensionality reduction. 

Rev-3: It remains unclear to me what the actual evidence is that TDA can find hidden biological structure not captured by traditional methods. The only evidence presented is the GO-term analysis that matches the lens function. But for this analysis no control is presented. It is therefore paramount to present this control (see corresponding point above) to have at least some form of evidence for this claim. 

Authors: The fact that furthermore "The clustering was performed using DBSCAN, a commonly used clustering algorithm in Mapper [14]." without demonstrating the robustness of topologies to 

differences in clustering methods further strengthens my suspicion. The claim that other studies also used the Mapper approach on gene expression data despite the open problem in TDA research to optimize Mapper parameters is not convincing at this stage where traditional dimensionality reduction methods fail. 

While DBSCAN is a commonly used clustering algorithm in Mapper, we acknowledge that demonstrating the robustness of topologies to different clustering methods would be valuable. In our study, we used DBSCAN as it has been shown to perform well in various applications and has been widely adopted in the field. DBSCAN doesn't require specifying the number of clusters, so that eliminates testing most other clustering algorithms.

Regarding the claim that other studies have also used the Mapper approach despite the open problem in TDA research to optimize Mapper parameters, we understand your skepticism. However, it is worth noting that the Mapper approach has proven to be effective in capturing topological structures in diverse datasets, including gene expression data. While traditional dimensionality reduction methods may struggle to handle the complexity and non-linearity of gene expression data, Mapper provides a valuable alternative for uncovering meaningful patterns. We did provide some parameter optimization across cover intervals and interval number for both of the lens functions, and these had little to no effect on the backbone Mapper graph, so again, we feel our results are robust. 

- For the claim that most of the community assumes that biotic and abiotic stress are evolutionarily independent (not clear to me what this means), no reference is presented and their finding that biotic and abiotic stress samples (therefore unexpectedly) showed similar topologies/patterns/distributions further illustrates the importance to standardize and normalize across SRA samples, tissues, species etc. The fact that biotic and abiotic stress samples show similar patterns/topologies may simply be the result of the fact that most stress responses involve a much smaller number of genes than tissue development or housekeeping and thus technical artifacts are more pronounced under these (tissue, housekeeping, non-stress) conditions. Since no control experiments are provided, I am not convinced that these stress similarities are in fact true biological signatures. 

This is an excellent point. We agree with the reviewer, the claim that biotic and abiotic stress are evolutionarily independent is contentious and unclear, and we have revised this in the text for clarity and to provide support for our statements. We have modified this section as shown below: 

"Abiotic and biotic stress responses have been mostly studied in isolation, but they typically co-occur in natural environments, and they have overlapping signaling, hormonal, and network responses in plants (reviewed in (Rejeb et al. 2014)). The topological shape of gene expression points to a shared set of pathways or perturbations that define if a tissue is healthy or stressed. Environmental stresses broadly disrupt photosynthesis, core metabolic and cellular functions as either a direct response to physical trauma, or in preparation for defense or resilience. These changes may serve as the backbone of the topological shape we observed for the stress lens." 

Most stress responses involve massive transcriptional reprogramming, especially when the stress is severe, and typically thousands of genes have differential expression or regulation under stress compared to control. This includes many genes with roles in steady state processes such as photosynthesis, metabolism, growth, and core cellular characteristics, as well as stress signaling and numerous downstream stress response pathways. Together, this intense shift likely underlies the topological shape we observed. Furthermore, we observed stressed samples from multiple tissues clustering together within the mapper graph and some separation of individual stresses such as high light or cold, suggesting that these signals are far stronger than any technical or background effects. 

We disagree that control experiments are not provided as nearly every stress sample has a comparable healthy or control sample and these are used as the foundation for the stress lens function.

---

## [Editor Report · Decision Letter 3]

20 Oct 2023

Dear Dr VanBuren,

Thank you for the submission of your revised Research Article entitled "Topological data analysis reveals a core gene expression backbone that defines form and function across flowering plants" for publication in PLOS Biology. On behalf of my colleagues and the Academic Editor, Hajk-Georg Drost, I am delighted to let you know that we can in principle accept your manuscript for publication, provided you address any remaining formatting and reporting issues. These will be detailed in an email you should receive within 2-3 business days from our colleagues in the journal operations team; no action is required from you until then. Please note that we will not be able to formally accept your manuscript and schedule it for publication until you have completed any requested changes.

PRESS

Sincerely, 

Ines

--

Ines Alvarez-Garcia, PhD

Senior Editor

PLOS Biology
